# Plasmon-Enhanced Perovskite Solar Cells Based on Inkjet-Printed Au Nanoparticles Embedded into TiO_2_ Microdot Arrays

**DOI:** 10.3390/nano13192675

**Published:** 2023-09-29

**Authors:** Sofia Rubtsov, Albina Musin, Viktor Danchuk, Mykola Shatalov, Neena Prasad, Michael Zinigrad, Lena Yadgarov

**Affiliations:** 1Department of Chemical Engineering, Biotechnology and Materials, Faculty of Engineering, Ariel University, Ariel 4076414, Israel; sofiar@ariel.ac.il (S.R.); viktorde@ariel.ac.il (V.D.); mykolash@ariel.ac.il (M.S.); neenaphy@gmail.com (N.P.); zinigradm@ariel.ac.il (M.Z.); 2Physics Department, Faculty of Natural Sciences, Ariel University, Ariel 4076414, Israel; albinam@ariel.ac.il

**Keywords:** perovskite solar cells, TiO_2_, plasmonic Au nanoparticles, inkjet printing, localized surface plasmon resonance

## Abstract

The exceptional property of plasmonic materials to localize light into sub-wavelength regimes has significant importance in various applications, especially in photovoltaics. In this study, we report the localized surface plasmon-enhanced perovskite solar cell (PSC) performance of plasmonic gold nanoparticles (AuNPs) embedded into a titanium oxide (TiO_2_) microdot array (MDA), which was deposited using the inkjet printing technique. The X-ray (XRD) analysis of MAPI (methyl ammonium lead iodide) perovskite films deposited on glass substrates with and without MDA revealed no destructive effect of MDA on the perovskite structure. Moreover, a 12% increase in the crystallite size of perovskite with MDA was registered. Scanning electron microscopy (SEM) and high-resolution transmission electron microscopy (HR-TEM) techniques revealed the morphology of the TiO_2__MDA and TiO_2_-AuNPs_MDA. The finite-difference time-domain (FDTD) simulation was employed to evaluate the absorption cross-sections and local field enhancement of AuNPs in the TiO_2_ and TiO_2_/MAPI surrounding media. Reflectance UV-Vis spectra of the samples comprising glass/TiO_2_ ETL/TiO_2__MDA (ETL—an electron transport layer) with and without AuNPs in TiO_2__MDA were studied, and the band gap (*E*_g_) values of MAPI have been calculated using the Kubelka–Munk equation. The MDA introduction did not influence the band gap value, which remained at ~1.6 eV for all the samples. The photovoltaic performance of the fabricated PSC with and without MDA and the corresponding key parameters of the solar cells have also been studied and discussed in detail. The findings indicated a significant power conversion efficiency improvement of over 47% in the PSCs with the introduction of the TiO_2_-AuNPs_MDA on the ETL/MAPI interface compared to the reference device. Our study demonstrates the significant enhancement achieved in halide PSC by utilizing AuNPs within a TiO_2__MDA. This approach holds great promise for advancing the efficiency and performance of photovoltaic devices.

## 1. Introduction

Hybrid organic–inorganic metal halide perovskites have excellent optoelectronic properties for photovoltaic applications owing to their adjustable band gap, low exciton binding energy, and high carrier mobility. This has made them an attractive area of research in simple-solution-based chemistry and in the search for accessible methods for depositing thin polycrystalline perovskite films at room temperature [1]. Perovskite-based solar cells (PSC) have shown rapid improvements in durability and efficiency in recent years. The last PSC has an efficiency of 25.8% [2], below the Shockley–Queisser limit (about 33.7%) [3] of the maximum theoretical efficiency for one-junction solar cells (SC), indicating untapped potential for further enhancing PSC device performance [4]. However, a common disadvantage of thin-film SCs is that only some of the incident light can be absorbed due to the small thickness of the active layer. Phillips et al., reported that the methyl ammonium lead iodide (MAPI) perovskite layer, measuring 350 nm in thickness, absorbed no more than 85% of incident light [5]. Also, Liu et.al. reported that as the thickness of a perovskite film surpassed the carrier diffusion lengths, there was a notable decline in the performance of PSCs, with the optimum thickness approximately at 330 nm [6]. So, increasing the absorption in the solar spectrum range and, as a result, gaining photoelectrons, is one of the ways to increase the efficiency of energy conversion. Various approaches and structural configurations can be employed to enhance absorption efficiency. For instance, creating a composite semiconductor oxide can lead to substantial improvements in power conversion efficiency [7]. Additionally, utilizing quantum dots can facilitate the conversion of photoelectrons, with the photoanode serving as a channel for electron transmission [8]. Incorporating organic and inorganic double-hole layers enables precise adjustment of energy level alignment between these layers, ultimately enhancing the photovoltaic performance and stability of PSCs [9]. Apart from this, another promising approach to enhance the capture of light in the submicron active layer of PSC is based on the introduction of metallic nanostructures in which localized surface plasmon resonance (LSPR) can be excited [10]. This leads to various effects that can occur in thin-film PSCs. First of all, metal nanoparticles (NPs) scatter light, increasing the effective path length within a layer and enhancing light absorption. Due to the LSPR, NPs function as subwavelength antennas, amplifying the near field and increasing the number of generated electron-hole pairs. Two more important effects are hot electron transfer, that is, the direct generation of charge carriers in the surrounding semiconductor material, and plasmon resonant energy transfer, both of which can significantly impact absorption efficiency [10,11].

Among the numerous studies of effects arising when plasmonic structures are coupled with perovskite materials [12,13], we can mention only a few works devoted to the application of spherical AuNPs in a TiO_2_ environment [14]. AuNPs with a diameter of 40 nm were incorporated between two TiO_2_ layers. The device with AuNPs showed 20–30% power conversion efficiency (PCE) enhancement compared to the planar perovskite device with a TiO_2_-fabricated ETL. The improved device performance was explained on the base of enhanced conductivity and decreased surface potential of TiO_2_ film, ascribed to the plasmon-mediated hot carrier injection from AuNPs to TiO_2_. The optimized PSC with 80 nm-sized Au@TiO_2_ nanospheres in mesoporous TiO_2_ and perovskite layers showed over 44% PCE enhancement compared to the reference device without the AuNPs [15]. The authors concluded that the enhanced device performance was due to plasmonic effects rather than only optical ones. A dual ETL with embedded AuNPs was developed [16]. Here, the bottom was a hybrid layer of plasmonic AuNPs (about 16 nm diameter) embedded in anatase TiO_2_ films, while the top layer was made of anatase TiO_2_ NPs to prevent direct contact between AuNPs and the perovskite. The PSCs with dual ETLs containing 0.2 wt% Au achieved power conversion efficiencies up to 20.31 and 15.36% better than those with TiO_2_ ETL only (on rigid and flexible substrates, respectively). In all these works, AuNPs were synthesized with the citrate method, and TiO_2_ layers were deposited by spin-coating.

In this work, we introduced a novel interface tailoring technique by inkjet printing TiO_2_ microdot arrays with embedded AuNPs (TiO_2_-AuNP_MDA) into a PSC as an additional layer. It has been demonstrated that the addition of the TiO_2_-AuNP_MDA on the PSC ETL layer results in the appearance of photoelectrically active regions, which both effectively utilize photon energy and manage electrons, increasing PSC efficiency [12,13,14]. However, upon direct contact with a perovskite absorber, AuNPs can act as recombination centers for charge carriers that degrade device performance [17]. The TiO_2_-AuNP_MDA layer inkjet printed at the ETL/MAPI interface can both isolate AuNPs from MAPI and create good contact with TiO_2_ ETL, ensuring the transfer of hot electrons and promoting photoconductivity and the extraction of photogenerated electrons at the TiO_2_/MAPI interface [18,19]. The printing of an MDA pattern rather than a continuous layer was chosen because individual droplets are reproducible, and AuNPs with the same size distribution were synthesized in them after annealing [20]. In addition, the MDA layer contributes to lower series resistance. The PSC modified with TiO_2_-AuNP_MDA demonstrated improved photovoltaic characteristics. Namely, the PCE of the champion cell was increased by more than 46% compared to a reference device without AuNPs. The proposed method will pave the way for more efficient PSCs.

## 2. Materials and Methods

Experimental information, including the methods, synthesis, PSC fabrication, and deposition of the layers, is provided in the Appendix A.

### 2.1. Solar Cell Device Structure

High absorbance and low interfacial resistance are critical for optimizing the PSC in the SC fabrication process. Herein, an increased absorbance is achieved by adding TiO_2_-AuNPs_MDA. At the same time, low interfacial resistance is attained by the microdot architecture of the additional layers. Here, we fabricate three types of PSCs of a planar architecture: the reference devices and their structure are presented in Figure 1A, modified PSCs with TiO_2__MDA on the ETL/MAPI interface are presented in Figure 1B, and PCS s with TiO_2_-AuNP_MDA on the ETL/MAPI interface are presented in Figure 1C. An energy level diagram of PCS with TiO_2_-AuNP_MDA is presented in Appendix A. The chosen MDA architecture of the additional layers promotes MAPI percolation, thus reducing the effective resistivity [21].

### 2.2. Characterization

The structural characteristics of the MAPI films deposited on bare glass, on glass with printed TiO_2__MDA, and on glass with TiO_2_-AuNP_MDA/MAPI surfaces were determined using an XRD (SmartLab SE, Rigaku, Akishima, Japan) powder diffractometer with Cu-Kα radiation (λ = 1.5460 Å). Measurements were made in 2θ geometry (angle of incidence 3°) in the range of 10–80° with a step of 0.03° and a rate of 0.5°/min. The phase analysis was performed using SmartLab Studio II ver. 4.2.44.0 with Powder XRD module and ICDD PDF-2 2019 database, using the Rietveld Refinement method in the range of 10–45° using the WPPF (Whole Powder Pattern Fitting) module. The crystallite size was calculated using the comprehensive analysis module and the Halder–Wagner modeling method. The morphology of TiO_2_-AuNP_MDA/MAPI films was studied with a TESCAN MAIA3 (Brno, Czech Republic) scanning electron microscope in SEM and STEM (scanning transmission electron microscope) modes. In detail, the compositional and crystallographic analysis of the single TiO_2_-AuNP dot was conducted with high-resolution TEM (HR-TEM) imaging. ImageJ software (LOCI, University of Wisconsin, Madison, WI, USA) with a modified “particle size” plugin was used to analyze the SEM micrographs in order to find the AuNP size distribution. The optical absorption spectra (recalculated from reflectance) of the prepared ETL/MDA films were measured with a spectrophotometer Sphere Jasco V-750 (JASCO Co., Tokyo, Japan). The current–voltage (I–V) characteristics of the fabricated PSCs were measured using a test station (Abet Technologies, Milford, CT, USA) with SunLite^TM^ solar simulator and a source meter Keithley 2400 (TEKTRONIX, Inc., Bracknell, UK) under 100 mW cm^−2^ illumination calibrated with a reference silicon PSC. The stainless steel mask with a 0.04 cm^2^ aperture was used to specify the illumination area of the PCS.

### 2.3. Simulations

Finite-difference time-domain (FDTD) simulation effectively evaluates the absorption cross-sections and local field of nanoscale optical devices. This study used commercially available Ansys Lumerical FDTD software (Ansys Inc., Canonsburg, PA, USA) to perform simulations that solve Maxwell’s equations in time and space. A Total-Field Scattered-Field (TFSF) was used as an excitation source to measure the local electric field response. The frequency domain field and power monitor directly measure the local field enhancement that the 2D monitor captured. Here, we measured the net power flowing into the particle and obtained the absorption cross-section (E/E^0^) by normalizing it to the source intensity (E^0^). Similarly, the scattering cross-section was calculated using an analysis group located outside the TFSF source [22,23]. This study modeled AuNPs embedded in TiO_2_ and MAPbI_3_ environments with “FDTD Mie scattering” as a reference model. The background index of the FDTD region was set to 1 for air, and the dielectric functions of MAPbI_3_, TiO_2_, and Au were taken from the previous reports [24,25]. The cross-section of the simulation scheme is presented in Appendix A (see Appendix A). Here, the AuNP diameter was set to 20 nm. For reference, we also simulate the optical response of AuNP, TiO_2_, and MAPbI_3_ surrounded by an environment of a constant refractive index (n_0_ = 1) [26].

## 3. Results and Discussion

### 3.1. XRD Studies

Structural characteristics are the determining factor for the physicochemical properties of materials. They are directly related to the processes of nucleation and growth, which in turn depend on the topology and properties of the substrate surface. Thus, the modification of the surface with MDA can affect the deposited MAPI layer. The structural characteristics of the MAPI films spin-coated on glass (G), on glass with printed TiO_2__MDA (GT), and with TiO_2_-AuNP_MDA (GTA) were determined with powder X-ray diffraction, and the corresponding patterns are presented in Figure 2.

The Rietveld analysis of the XRD data revealed that the MAPI films were polycrystalline and consisted of two phases: tetragonal MAPI described with an *I422* space group of symmetry and a minor amount of trigonal PbI_2_ phase *R-3m*. Modifying the glass surface with TiO_2__MDA increases the PbI_2_ phase content from 7.5 ± 0.4 to 13.6 ± 0.3 w%. The TiO_2_-AuNP_MDA has a more negligible effect on the phase composition, increasing the PbI_2_ content only to 9.9 ± 0.3 w%. A similar trend also persists for unit cell volumes of MAPI deposited on G, GT, and GTA substrates, which correspondingly 992 Å^3^, 996 Å^3^, and 995 Å^3^. Still, it is interesting to note that the presence of MDA on the glass surface increases the coherent scattering regions of the MAPI films from 286 ± 8 to 308 ± 4 and 321 ± 5 Å correspondingly for G, GT, and GTA substrates. Thus, the MDA printing positively affects the crystallite’s size of the MAPI film with minor changes in phase composition and unit cell volume. Note that the X-ray analysis did not reveal any reflections characteristic of the titanium oxide and gold phases from the corresponding MDA; the MAPI film completely hides these peaks. Moreover, XRD analysis demonstrates consistent intensity levels across all samples, whether with or without TiO_2__MDA and TiO_2__AuNP MDA. This provides a direct verification that the application of MDA does not affect the thickness of the MAPI layer.

### 3.2. Morphology Analysis

The SEM images of the TiO_2_-AuNP_MDA on the TiO_2_ ETL after annealing are presented in Figure 3A. It was revealed that the spacing between the drops was about 80 µm, and their average diameter was around 35 µm (see Figure 3B). The average thickness of the MDA is about 35 ± 10 nm. (The detailed calculation is presented in the Appendix A). Figure 3C illustrates the presence and arrangement of AuNPs in a TiO_2_ matrix. The size distribution of AuNPs was calculated with “ImageJ” software to be in the range of 25 ± 14 nm and presented in Appendix A.

The HR-TEM image in Figure 3D revealed the crystalline planes of Au and TiO_2_ in the TiO_2_/AuNP dot printed on an amorphous carbon-coated Cu grid and heated to 200 °C to remove solvents. The inter-planar spacing of 0.23 nm (yellow-marked in Figure 3D) corresponds to the (111) plane of Au in the FCC phase [27], while the blue spacing of 0.33 nm can be assigned to rutile TiO_2_ reflection (110) [28]. A space plane of TiO_2_ was found on the top of the AuNPs, demonstrating that AuNPs are indeed covered with TiO_2_, Appendix A. The planes with the same spacing of 0.33 nm can also be found on the TiO_2_ material surrounding the AuNP, which confirms the identification of TiO_2_ on the AuNP’s surface. This is a significant fact since the surface of the metal NP should not come into contact with MAPI to avoid degradation. Covering AuNP with TiO_2_ is an essential process that minimizes recombination on metallic NPs and improves charge carrier separation [29].

### 3.3. Grain Size Analysis of MAPI

One of the parameters that influence the photovoltaic performance of PSCs is the MAPI grains’ size [30]. SEM imaging of the MAPI film surface and analyses of grains size distribution using ImageJ software have been performed to examine the effect of MDA on the morphology of the perovskite layer. The SEM images of the MAPI film surface are presented in Figure 4. The grain diameters of MAPI on a bare glass substrate, glass with printed TiO_2__MDA, and glass with TiO_2_-AuNP_MDA were found to be 128 ± 72, 142 ± 110, and 125 ± 68 nm, respectively. From the SEM analysis, the TiO_2__MDA does not influence the uniformity of the MAPI film. According to the SEM micrograph of the MAPI samples’ cross-section, the thickness of the latter is about 400 nm. (Appendix A). The low-magnification SEM images of all the films are shown in Appendix A. The macroscale uniformity and smoothness of all the examined films can be readily observed.

### 3.4. FDTD-Simulated Electric Field Response

According to previous reports, AuNPs can improve the efficiency of PSCs by enhancing light absorption, accelerating exciton separation, promoting carrier extraction, and enabling hot electron injection [10,11,12,13,14,15,16]. Specifically, the AuNPs can act as “antennas” using the LSPR effect to concentrate the incident light and achieve a “light-trapping” effect [31]. This near-field enhancement can act as a supplementary light source, thus enhancing absorption [13]. The extent of the plasmonic enhancement in PSCs not only depends on the LSPR properties but is also closely related to the structure of PSCs. Thus, evaluating the enhancement extent within the dielectric environment of PSCs is of the utmost importance [31,32]. Here, we use the FDTD simulation to examine the near-field absorbance of the bare Au NP and compare it to NPs embedded in TiO_2_ and MAPI, see Figure 5. The simulation results revealed a localized enhancement of the electric field in the TiO_2_/AuNP structure at a wavelength of 618 nm, Figure 5B. In contrast, the TiO_2_/AuNP-MAPI configuration exhibited its peak enhancement at a wavelength of 630 nm, as illustrated in Figure 5F. Combining Au NPs with a dielectric material is expected to further enhance the LSPR effect by increasing the confinement of the electromagnetic field near the NPs’ surface [14,33,34]. Indeed, the electric field of the Au NP surrounded by TiO_2_ presented in Figure 5B is significantly higher than that of its bare counterpart (Au NP in the air). In fact, the derived electric-field (|E^2^|) intensity of Au NPs embedded in TiO_2_ and TiO_2_/MAPI was found to be increased fourfold and eightfold, respectively. The near-field enhancement of the electromagnetic field in the simulated AuNP/TiO_2_/MAPI hybrid structure (see Figure 5F) is higher due to the strong excitonic absorption of the MAPbI_3_.

### 3.5. Simulated and Estimated Absorption Using FDTD and Reflectance Measurements

Absorption cross-sections of AuNP, TiO_2_, MAPI, and their hybrids were extracted from the FDTD simulations. Figure 6A presents the simulated absorption cross-section of the AuNP in different dielectric media. We find that the simulated LSPR of the AuNP suspended in air is at ~508 nm (Appendix A), corresponding to the previous reports and experimental measurements [35,36]. The LSPR is red-shifted from 508 nm to 618 nm and 630 nm as the dielectric environment of the AuNP is changed to be TiO_2_ and TiO_2_/MAPI, respectively. The Mie theory can explain the shift of the LSPR maxima [37,38,39]. Indeed, similar findings were also observed when the relationship between the LSPR extinction band of spherical AuNPs was analyzed using discrete dipole approximation theory and the Drude model [40].

The intensity of the shifted LSPR is increased by 12 and 10 times when the AuNP was embedded in TiO_2_ and MAPI, respectively. Non-normalized curves of the absorption cross-sections are presented in Appendix A. Moreover, the simulated absorption of MAPI/TiO_2_ vs. MAPI/TiO_2_/AuNP hybrids indicates an increase of ~27 times at 630 nm. The phenomena of the increased LSPR intensity can be assigned to the stronger confinement of the electromagnetic field near the NP’s surface [14,33,34].

### 3.6. Optical Properties

UV-Vis absorbance (calculated from the reflectance, (Appendix A) spectra of all the examined samples are shown in Figure 6B. Contrary to the simulation results, the measured spectra do not exhibit a significant increase in absorbance intensity. That discrepancy between the simulated and experimental absorbance can be explained by the perfect conditions of the simulation vs. the real ones. Namely, we simulated one AuNP, whereas the experimental setup consists of multiple NPs with broad size distribution.

While we can clearly see the impact of AuNP in the simulation, the experimental absorbance does not exhibit a similarly remarkable improvement. (Figure 6A,B). This discrepancy is mainly rooted in the ideal simulation environment where we focused on a single NP. Notwithstanding, the absorbance of AuNPs in the examined samples is challenging to detect experimentally because of the thick layer of MAPI. However, the spectra exhibit an observable difference between the MAPI/TiO_2_-AuNP_MDA and MAPI/TiO_2__MDA in the 615–690 nm range. (Figure 6B). Indeed, there is an increase in the absorbance intensity of the samples that contain TiO_2_-AuNP_MDA. The range of the enhanced absorbance corresponds with the LSPR of AuNPs and thus can be ascribed to the presence of the AuNPs in the MAPI/TiO_2_-AuNP_MDA. Further, the band gap (*E*_g_) values from the reflectance spectra were calculated using the Kubelka–Munk equation, and it was found to be around 1.6 eV for all samples covered with MAPI (see Appendix A). This observation is in good agreement with the known *E*_g_ value (1.61 eV) for the MAPI deposited on mesoporous TiO_2_. Notably, the introduction of MDA had no impact on the band gap value, which consistently remained at approximately 1.6 eV across all samples.

### 3.7. Photovoltaic Parameters of PSCs

To reveal the LSPR enhancement of the PCS performance, the TiO_2_-AuNP_MDA was introduced at the ETL-MAPI interface. Three types of PSC devices were fabricated: type 1—reference device, type 2—with TiO_2__MDA, and type 3—with TiO_2_-AuNP_MDA. The characteristics of the measured PCS devices are shown in Table 1. The comparison of the performance of previously reported devices comprising AuNPs in perovskite solar cells is presented in Appendix A. Short-circuit photocurrent (*J*_SC_), open-circuit voltage (*V*_OC_), power conversion efficiency (PCE), and fill factor (FF) are the key parameters of the performance of SCs and were obtained from the analysis of current-voltage (I–V) characteristics of the SCs using LabView-based homemade program, PSC fabrication section is presented in Appendix A.

The data presented in Table 1 indicate that the presence of the TiO_2__MDA improves the PCE and the FF. The possible explanation could be the MAPI scaling effect, i.e., the TiO_2__MDA will have a higher ETL surface contact area with MAPI [23]. Moreover, the MDA architecture does not increase the effective resistance of the PSC, and the *J*_SC_/*V*_OC_ remains similar to the reference sample. It is important to note that if a continuous layer of TiO_2_-AuNPs is introduced into the MAPI/ETL interface, we would expect a prominent decay in the *V*_OC_ parameter [14]. The same samples were evaluated after a week of storage, see Appendix A. The TiO_2_-AuNP_MDA sample shows a slight improvement of *J*_SC_ parameter. Adding TiO_2_-AuNP_MDA to the PSC further increases measured parameters, resulting in a ~47% enhancement in PCE compared to the reference device. Statistical analysis of the optimized PSCs’ parameters, such as *J*_SC_, *V*_OC_, PCE, and FF, for all the tested devices and their average values are presented as a box plot shown in Figure 7. Nonetheless, it remains challenging to assert that the performance of PSCs, especially in terms of conversion efficiency, has solely improved due to the presence of Au NPs, necessitating further investigation. However, it is noteworthy that even the average values illustrate an improvement in PSCs following the integration of TiO_2__AuNP MDA in comparison to reference devices. The corresponding calculated increase is depicted in Figure 7.

The schemes clearly demonstrate that the device with the TiO_2_-AuNP_MDA structures has higher *J*_SC_, PCE, and FF values than the reference cells. In general, the reason for improving the photovoltaic parameters of PSCs with dot arrays may be due to the increased loading of the MAPI absorber in the porous structure of TiO_2_ dots. However, according to the statistical analysis, the samples with TiO_2__MDA do not exhibit an increase in current density. Hence, the improved PSC features might be attributed solely to the enhanced absorption accompanied by an increased light scattering within a cell induced by AuNPs. Moreover, according to the XRD data, embedding the TiO_2__MDA or TiO_2_-AuNP_MDA on an ETL/MAPI interface increases the perovskite crystallite size, thus having a positive effect on FF. Finally, the FF and *J*_SC_ enhancement due to TiO_2_-AuNP_MDA introduced on the ETL/MAPI interface led to around 47% PCE improvement of the PSC. The LSPR approach is a prospective way to increase PSC efficiency.

## 4. Conclusions

In summary, this work highlights a novel and efficient strategy for the inkjet-printing fabrication of high-performance plasmonic architectures based on 2D arrays of TiO_2_ microdots with embedded AuNPs, and presents the superiority of the plasmon-enhanced perovskite solar cell device characteristics. The XRD analysis confirmed the perovskite structure of the photoactive MAPI layer synthesized on top of the microdot array. Moreover, the presence of the microdot array increases the crystallite size of MAPI. Important to note is that HR-TEM images revealed crystalline planes of TiO_2_ on the top of the AuNPs, demonstrating that AuNPs were indeed embedded into TiO_2_ microdots. In general, incorporating AuNPs in the TiO_2_ matrix improves light absorption in the visible region [13]. However, we unambiguously observed an enhanced photocurrent without significant variation in the light-harvesting capability in this study. Thus, the enhancement of the internal conversion efficiency of the fabricated perovskite solar cell with Au incorporation can be either due to enhanced charge separation/free carrier generation or due to enhanced charge collection. Herein, the origin of enhanced photocurrent is attributed to a previously unobserved and unexpected mechanism of reduced exciton binding energy with the incorporation of the metal NPs rather than enhanced light absorption [17]. Henceforth, the inkjet printing of 2D microdot arrays of dielectric material with incorporated plasmonic NPs is a promising approach to increase the efficiency of photovoltaic devices. The demonstrated enhancement of perovskite solar cell efficiency through AuNP printed microdots offers a glimpse into the exciting prospects of achieving more efficient and sustainable energy generation, thus contributing to the ongoing global efforts towards a greener and cleaner energy future. 

## Figures and Tables

**Figure 1 nanomaterials-13-02675-f001:**
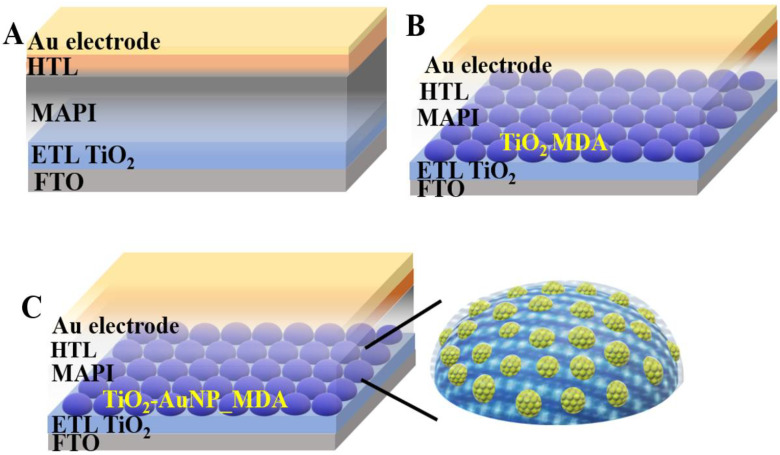
Scheme of the PSC devices: (**A**)—standard perovskite PSC with planar architecture, (**B**)—PSC with TiO_2__MDA, and (**C**)—PSC with TiO_2_-AuNP_MDA.

**Figure 2 nanomaterials-13-02675-f002:**
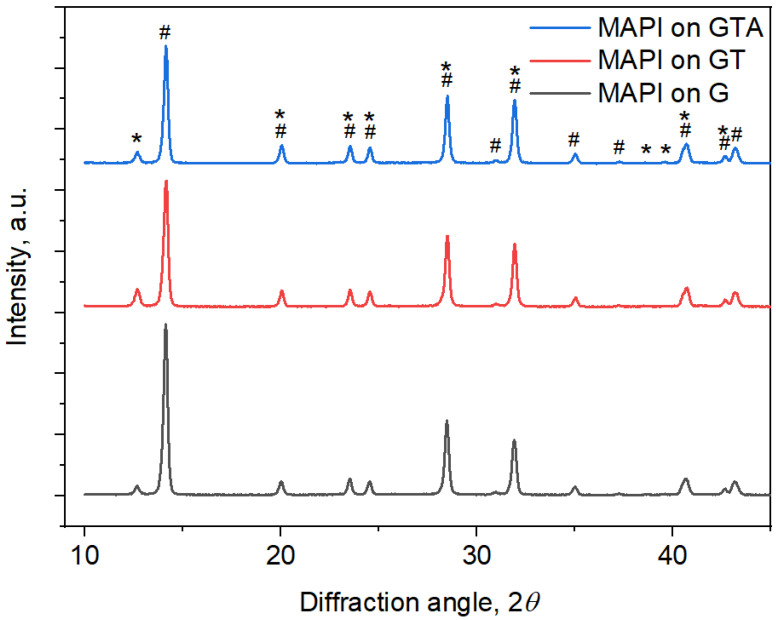
The X-ray diffraction patterns of the MAPI layers synthesized on a glass (G), on a glass with TiO_2__MDA (GT), and on a glass with TiO_2_-AuNP_MDA (GTA), represented by black, red, and blue lines, respectively. The MAPI phase peaks are denoted with the **#** symbol and the PbI_2_ reflections with the ***** one.

**Figure 3 nanomaterials-13-02675-f003:**
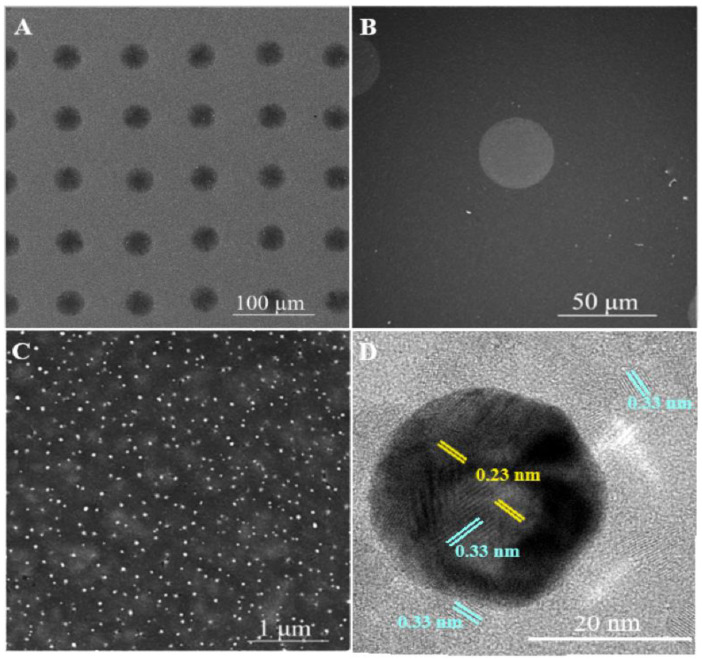
SEM images of (**A**) TiO_2_-AuNP_MDA printed on ETL; (**B**) a single TiO_2_-AuNP_MDA; (**C**) AuNPs in TiO_2__MDA; (**D**)—HR-TEM micrograph of an Au NP embedded in TiO_2_. Yellow mark—the inter-planar spacing of FCC face of Au; blue mark—the inter-planar spacing of rutile TiO_2_.

**Figure 4 nanomaterials-13-02675-f004:**
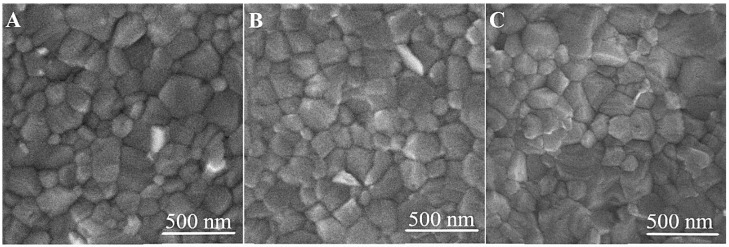
SEM images of MAPI surface on (**A**) glass substrates, (**B**) glass with TiO_2__MDA, and (**C**) glass with Au/TiO_2__MDA.

**Figure 5 nanomaterials-13-02675-f005:**
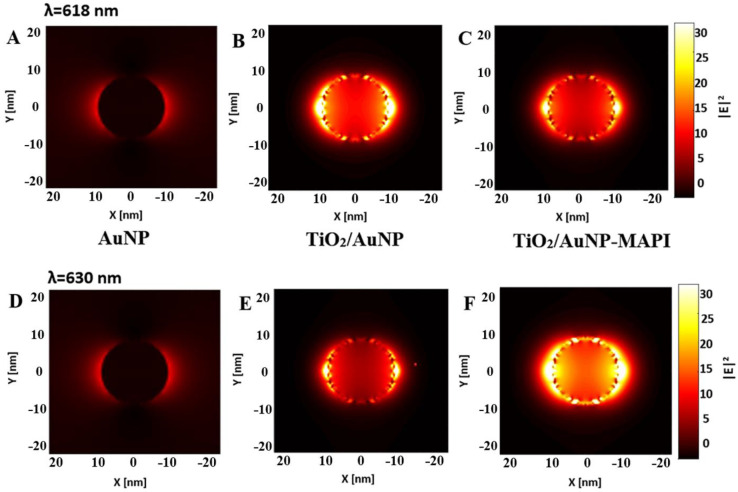
FDTD simulated electromagnetic intensity distribution under 618 nm wavelength: of (**A**) AuNP, (**B**) TiO_2_/AuNP, and (**C**) TiO_2_/AuNP/MAPI; under 630 nm wavelength: of (**D**) AuNP, (**E**) TiO_2_/AuNP, and (**F**) TiO_2_/AuNP/MAPI. The E^2^ here is collected according to the absorbance maxima shown in Appendix A. The magnitude of the enhanced electric field intensity is indicated by the color scale on the right.

**Figure 6 nanomaterials-13-02675-f006:**
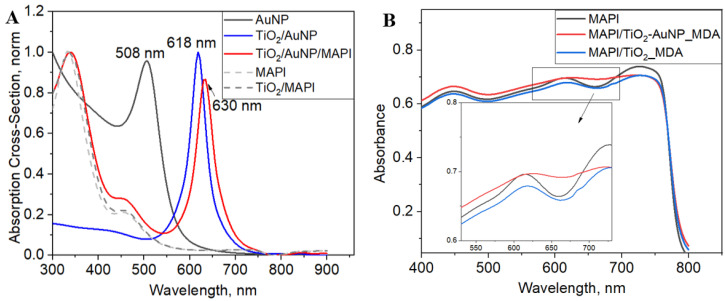
(**A**) FDTD-simulated normalized absorption cross-section; (**B**) absorbance of MAPI, MAPI/TiO_2__MDA, and MAPI/TiO_2_-AuNP_MDA samples.

**Figure 7 nanomaterials-13-02675-f007:**
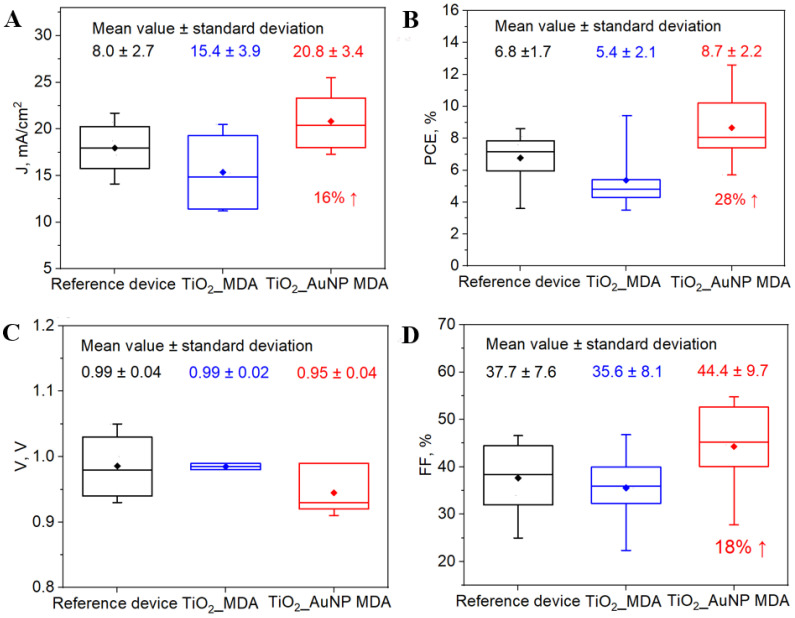
Comparison of the average values of (**A**) *J*_SC_, (**B**) PCE, (**C**) *V*_OC_, and (**D**) FF of the fabricated PCS.

**Table 1 nanomaterials-13-02675-t001:** Photovoltaic performance of the champion devices.

Sample Code	*J*_SC_ mA/cm^2^, ±0.1	*V*_OC_ V, ±0.01	PCE %, ±0.3	F.F. %, ±0.1
1 Reference device	19.8	0.98	8.6	44.4
2 with TiO_2__MDA	20.5	0.98	9.4	46.8
3 with TiO_2_-AuNP_MDA	23.3	0.99	12.6	54.8

## Data Availability

Not applicable.

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
