# Peer review of "Plasmon-Enhanced Perovskite Solar Cells Based on Inkjet-Printed Au Nanoparticles Embedded into TiO2 Microdot Arrays"

_nanomaterials, 2023, doi:10.3390/nano13192675_

Round 1

Reviewer 1 Report

In this study, the authors reported the local surface plasmon enhanced performance of perovskite solar cells (PSCs) embedded in titanium oxide (TiO2) micropoint arrays (MDA) deposited by inkjet printing technology using plasma gold nanoparticles (AuNPs). Relevant research has demonstrated that by using TiO2_ Significant enhancement achieved in halide PSC using AuNPs in MDA. This method is expected to improve the efficiency and performance of photovoltaic devices. I believe that publication of the manuscript may be considered only after the following issues have been resolved.

1.    In order to better reflect the physical mechanism of the device, the author needs to provide an energy band diagram of charge carrier transport under sunlight.

2.    In order to better demonstrate the superiority of the performance of device, it is recommended that the author provide a table to compare the reported work.

3.    The clarity of some images is insufficient, and the author needs to make adjustments, as shown in Figure 6

4.    The introduction can be improved. The author needs to mention some of the latest work on solar cells, such as Solar Energy 262 (2023) 111796; Dalton Transactions, 2023, 52, 81-89; Applied Thermal Engineering 230 (2023) 120841; Electrochimica Acta 412 (2022) 140145.

5.    The English expression of the whole article needs to be further improved.

Minor editing of English language required

Author Response

We would like to thank the reviewer for the constructive and very valuable comments. We have revised the manuscript and addressed all the queries of the reviewer. Please find the attached response to the reviewer's comment file.
thank you.

Reviewer 2 Report

This paper reports the results of an attempt to improve the efficiency of perovskite solar cells by TiO2 layer with Au nanoparticles fabricated by inkjet method. The patterned Au nanoparticles are expected to have optical effects such as plasmon, which can enhance the optical properties. As a result, an increase in the photocurrent of solar cells is expected. The authors fabricated a perovskite solar cell on a substrate with Au nanoparticles embedded into TiO2 microdots array, and verified its effect to the energy conversion efficiency of a perovskite solar cells. However, it is difficult to prove the optical effect of Au nanoparticles from the results of this paper alone. Therefore, I think this paper should be revised before it can be accepted by this journal.

Some comments are listed below.

1.

Fig 4 shows the SEM images of the surface. What do the cross-sectional SEM images look like? Does the Au nanoparticles affect the thickness of the perovskite film?

2.

Fig. 6A seems to show the normalized spectra. How can the authors explain the relationship between the emission of Au nanoparticles and the absorption spectra in Fig.6B?

3.

As shown in Fig.7, the conversion efficiencies are low and it is not clear whether the effect of Au nanoparticles is shown. I think it is difficult to explain that the conversion efficiency has increased due to the Au nanoparticles from these result, because their low efficiencies and large experimental errors.

4.

The proof of the formation of MAPbI3 crystals by XRD measurement and the simulation results on the optical effect of Au nanoparticles are very helpful. However, I do not think they are results that prove that they are linked to the improvement of perovskite solar cell efficiency. Therefore, I think it would be better to divide this paper into separate themes and summarize them.

Author Response

(The authors gave the same response as above.)

Round 2

Reviewer 1 Report

 Accept in present form